# Design and Activity of Novel Oxadiazole Based Compounds That Target Poly(ADP-ribose) Polymerase

**DOI:** 10.3390/molecules27030703

**Published:** 2022-01-21

**Authors:** Divakar Vishwanath, Swamy S. Girimanchanaika, Dukanya Dukanya, Shobith Rangappa, Ji-Rui Yang, Vijay Pandey, Peter E. Lobie, Basappa Basappa

**Affiliations:** 1Laboratory of Chemical Biology, Department of Studies in Organic Chemistry, University of Mysore, Mysore 570006, India; divakardivi166@gmail.com (D.V.); swamynayak010@gmail.com (S.S.G.); dukanya4@gmail.com (D.D.); 2Adichunchanagiri Institute for Molecular Medicine, Mandya 571448, India; shobithrangappa@gmail.com; 3Tsinghua Berkeley Shenzhen Institute, Tsinghua Shenzhen International Graduate School, Tsinghua University, Shenzhen 518055, China; yang.jirui@sz.tsinghua.edu.cn (J.-R.Y.); vijay.pandey@sz.tsinghua.edu.cn (V.P.); 4Institute of Biopharmaceutical and Health Engineering, Tsinghua Shenzhen International Graduate School, Tsinghua University, Shenzhen 518055, China; 5Shenzhen Bay Laboratory, Shenzhen 518055, China

**Keywords:** oxadiazole, poly(ADP-ribose) polymerase, human breast cancer, nicotinamide

## Abstract

Novel PARP inhibitors with selective mode-of-action have been approved for clinical use. Herein, oxadiazole based ligands that are predicted to target PARP-1 have been synthesized and screened for the loss of cell viability in mammary carcinoma cells, wherein seven compounds were observed to possess significant IC_50_ values in the range of 1.4 to 25 µM. Furthermore, compound **5u,** inhibited the viability of MCF-7 cells with an IC_50_ value of 1.4µM, when compared to Olaparib (IC_50_ = 3.2 µM). Compound **5s** also decreased cell viability in MCF-7 and MDA-MB-231 cells with IC_50_ values of 15.3 and 19.2 µM, respectively. Treatment of MCF-7 cells with compounds **5u** and **5s** produced PARP cleavage, H2AX phosphorylation and CASPASE-3 activation comparable to that observed with Olaparib. Compounds **5u** and **5s** also decreased foci-formation and 3D Matrigel growth of MCF-7 cells equivalent to or greater than that observed with Olaparib. Finally, in silico analysis demonstrated binding of compound **5s** towardsthe catalytic site of PARP-1, indicating that these novel oxadiazoles synthesized herein may serve as exemplars for the development of new therapeutics in cancer.

## 1. Introduction

Loss of function mutation of breast cancer genes 1/2 (BRCA1 and BRCA2) significantly increases the risk of development and progression of breast and other cancers [1]. Pre-clinical investigations determined that dysfunction of BRCA1/2 results in sensitization of cancer cells to the catalytic inhibition of Poly ADP-ribose polymerase (PARP), producing cell-cycle arrest followed by apoptosis [2]. This vulnerability (synthetic lethality) initially allowed pharmacological inhibition of PARP to be exploited in BRCA1/2 deficient cancer, which is now being extended to BRCA proficient cancer [3,4,5].

PARP1 is one member of a large family of structurally related proteins. PARP1 is a 113 kDa protein, which possesses three functional domains, namely the catalytic domain (CD), DNA-binding domain (DBD), and auto-modification domain (AMD) [6]. PARP-1 catalytically converts nicotinamide adenine dinucleotide (NAD^+^) into niacinamide and ADP-ribose, resulting in the transfer of ADP-ribose to downstream proteins modulating cellular functions. such as chromatin structure, gene transcription, DNA repair, and apoptosis [7]. Substrate mimetic compounds such as nicotinamide and 3-aminobenzamide were reported as weak PARP1 inhibitors with poor specificity, hence prompting chemists to design potent and specific PARP-1 inhibitors for clinical use [8,9,10]. To date, a number of PARP inhibitors (PARPis) with differing PARP trapping capacity have been clinically approved or received FDA Orphan Drug designation (Veliparib), including Olaparib, Talazoparib, and Rucaparib (Figure 1) [11].

However, both pre-clinical studies and clinical data demonstrate either intrinsic resistance to PARP inhibition or the development of acquired resistance to PARPi, resulting in limitation of the clinical response to the currently approved PARPis [12,13,14,15].

Replacement of the unsaturated heterocycles of many PARP inhibitors, such as niraparib, veliparib, E7016, CEP9722, and INO1001, with the saturated cyclic amine group generated a new generation of compounds that are reported to have low nano-molar inhibition of PARP activity and also display robust cellular activity [16,17,18]. In particular, compound 1 (Figure 1) bearing an oxadiazole moiety was observed to inhibit PARP1 activity with an IC_50_ as low as 1 nM resulting in a cellular EC_50_ of 3.7 nM; this was combined with an excellent pharmacokinetic profile [19]. Additionally, an oxadiazole containing a molecule called G007-LK (Figure 1) was reported as a potent inhibitor of tankyrases (TNKS) 1 and 2. G007-LK was co-crystallized with the PARP catalytic domain of TNKS2 and bound to the extended adenosine binding pocket of PARP, indicative of unique oxadiazole binding to PARP catalytic domains [20]. Recently, compound 2 (Figure 1) was also reported to inhibit the autophagy activating kinase 1 (ULK1) with an IC_50_ of 14 nM, and ULK1/2 inhibition in combination with Olaparib [21]. In continuation of the effort in the synthesis of novel oxadiazoles [22,23,24,25,26,27,28], herein synthesis of oxadiazoles with substitutions of pyridine, biphenyl, pyrimidine, and thiophene rings that target PARP1 and the efficacy of these compounds in BRCA wild-type estrogen receptor positive (ER+) and triple-negative breast cancer (TNBC) cells is reported.

## 2. Results and Discussion

### 2.1. Chemical Synthesis of Newer Oxadiazoles

3-bromobenzohydrazide **3a** synthesis was undertaken by refluxing ethyl 3-bromobenzoate **2a** and hydrazine, which was further reacted with different benzoic acids via intermolecular cyclization reaction using POCl_3_ as a cyclizing agent, and various oxadiazoles **4a**–**c** were obtained. The resulting oxadiazoles **4a**–**c** containing bromine were subjected to Suzuki coupling reactions with diverse boronic acids to obtain title compounds **5a**–**x** (Figure 1, Table 1). The synthesized oxadiazoles were characterized by ^1^H NMR, ^13^C NMR, LC-MS, HPLC, and melting points (Table 2 & refer to the Appendix A).

### 2.2. Efficacy of Oxadiazoles in Breast Cancer Cells

Initially, the newly synthesized oxadiazoles were examined in cell viability studies in human breast cancer (MCF-7) cells using Alamar Blue assays [29,30]. As internal control, Tamoxifen and Doxorubicin produced loss of viability of MCF-7 cells with IC_50_ values of 1.69 and 0.24 µM, respectively. The results of the assay revealed that compounds such as **5h**, **5m**, **5p**, **5r**, **5s**, **5u**, and **5w** inhibited the viability of MCF-7 cells with varied IC_50_ values of 21.8, 27.1, 18.4, 25.0, 15.3, 1.4, and 23.1 µM, respectively. All of the compounds that bear a pyridine ring produced significant loss of viability of MCF-7 cells. The compound **5u** exhibited IC_50_ values of 1.4 and 29.3 µM against MCF-7 and MDA-MB-231 cells, respectively, which was significant when compared to Olaparib (3.2 and 70.2 µM). The better activity of the compound **5u** could be attributed to the presence of the naked benzaldehyde group. One more compound, **5s**,was found to be effective againstboth MCF-7 and MDA-MB-231 cells where IC_50_ values of 15.3 and 19.2 µM, respectively (Figure 2). In addition to the lipophilic structures of these tested compounds, the presence of polar and active groups in the lead structures was found to decrease viability in human breast cancer cells.

### 2.3. Newer Oxadiazoles Inhibited the Catalytical Activity of PARP1 In Vitro

The most active oxadiazoles (**5u** and **5s**) were tested for their capacity to inhibit PARP1 activity using the PARP/Apoptosis Universal Colorimetric Assay Kit, as described [31,32]. The screening of these compounds was performed at concentrations of 0.01, 0.1, 1, 10, and 100 µM in triplicate (*n* = 3) measurements, in which the compound **5u** showed 4.82, 6.84, 45.39, 54.18, and 82.25% inhibition and **5s** showed 2.38, 10.58, 21.33, 64.99, and 76.50% inhibition, respectively. The positive control, 3-Aminobenzamide, inhibited PARP1 activity by 0.50, 6.17, 23.32, 56.63, and 85.74% at the tested concentrations, respectively (Figure 3).

### 2.4. Compounds ***5u*** and ***5s*** Increased PARP1 Cleavage, Phospho-H2AX Levels and CPP32 (Caspase-3) Activation in Breast Cancer Cells

As 1,3,4-oxadiazoles have been previously reported to cleave endogenous PARP1, the effects of compounds **5u**, **5s**, and Olaparib on PARP1 cleavage in MCF-7 and MDA-MB-231 cells were examined [33]. For this purpose, MCF-7 (Figure 4A) and MDA-MB-231 (Figure 4B) cells were incubated with compounds **5u** or **5s** or Olaparib at equivalent concentrations. Equivalent levels of cleaved PARP1 were observed in both cell lines after treatment with compounds **5u** or **5s** or Olaparib.

Furthermore, phosphorylation of the histone variant H2AX on Ser139, forming pH2AX, is an early response of a cell to DNA double-strand breaks [34]. Stress-induced DNA damage results in rapid phosphorylation of Ser139 of H2AX by PI3K-like kinases such as DNA-PK, ATM, and ATR [35]. Olaparib treatment has also been reported to significantly increase the level of p-H2AX [36,37]. Treatment of MCF7 and MDA-MB-231 cells with compound **5u** or **5s** or Olaparib produced equivalent increases in the level of p-H2AX in both cell lines (Figure 4). CPP32 (CASPASE-3) is a prototypical caspase that is activated during apoptosis [38,39,40]. CPP32 activity was measured after treatment of both cell lines with compounds **5u** or **5s** or Olaparib (Figure 4C,D). Again, similar results were observed with all three compounds in both cell lines.

### 2.5. Compounds ***5u*** and ***5s*** Inhibit Oncogenicity and Decreases 3D Growth of Breast Cancer Cells

MCF-7 and MDA-MB-231 cells were first cultured in monolayers in 24-well plates for foci formation assays and treated with compounds **5u**, **5s**, or Olaparib. The foci formed by MCF-7 (Figure 5A) and MDA-MB-231 (Figure 5B) cells were decreased in a dose-dependent manner, approximately equivalently, by all three compounds. Hence, compounds **5u** and **5s** inhibit the oncogenicity of both ER+ and triple-negative breast cancer cells.

To examine the efficacy of these compounds in an ex vivo organoid model, we grew both MCF-7 and MDA-MB-231 cells in 3D Matrigel^®^. Again, compounds **5u** and **5s** exhibited roughly equivalent efficacy to Olaparib in inhibiting 3D growth of both cancer cell lines (Figure 6).

### 2.6. In Silico Analysis of the Interaction of Compound ***5s*** with the PARP1 Catalytic Domain

To understand the interaction of the novel oxadiazoles with the PARP1 catalytic domain, the co-crystal structure of compound **33** [(9aR)-1-[(1-{2-fluoro-5-[(4-oxo-3,4-dihydrophthalazin-1-yl)methyl]benzoyl}piperidin-4-yl)carbonyl]-1,2,3,8,9,9a-hexahydro-7H-benzo[de][1,7]naphthyridin-7-one] with the PARP1 catalytic domain (PDB ID: 4HHY) was analyzed [41]. Accelrys DS version 2.5 was used for the molecular docking utilizing the previously reported protocol [42,43]. Compound **33** and the active oxadiazole compound **5s** were docked towards the catalytic domain of PARP1. The analysis of the results indicated that the core region of compound **33** called benzo[de][1,7]naphthyridin-7-one was occupying the active site of the catalytic domain of PARP1, which was similar when compared to the phenylic-pyridine group of active compound **5s**, which showed high negative CDOCKER energy. The co-crystal ligand compound **33** appears to possess higher negative CDOCKER energy of 45.8 kcal/mole, when compared to the most active compound **5s**, whose high negative CDOCKER energy was recorded as 28.1 kcal/mole. The detailed molecular interactions between the amino acid residues of the PARP1 catalytic domain and compound **33** or compound **5s** are shown in Figure 7. The detailed molecular interactions of compound **5s**, which exhibited higher binding affinities with the PARP1 catalytic domain by having stable H-bonds with Ser243, were also influenced by three intramolecular H-bonds between His201-Ala237, Asp105-Arg217, and Ser203-Asn207. p-p stacking of compound **5s** with Tyr228, Tyr235, Phe236, His201, and Tyr246 by fluoro, a methyl-substituted pyridinyl-phenyl group contributed to the net and higher negative CDOCKER energy of the ligand and enzyme complex.

## 3. Materials and Methods

All chemicals and solvents were purchased from Sigma-Aldrich (Bangalore, India). The completion of reactions was monitored by pre-coated silica gel TLC plates and hexane/ethyl acetate used as an eluent. ^1^H and ^13^C NMR were recorded on an Agilent NMR spectrophotometer (400 MHz); TMS was used as an internal standard and CDCl_3_ and DMSO were used as solvents; chemical shifts are expressed as ppm. Mass spectra and HPLC were collected on Waters (Xevo G2-XS QTof) (Wilmslow, UK) and Acquity UPLC (waters column: Sunfire C18, 4.6 × 250 mm) (Milford, MA, USA), respectively.

### 3.1. General Procedure for the Synthesis of 2,5-Disubstituted-1,3,4-oxadiazole

A mixture of benzoic acid (**1a**) (1 mmol), and concentrated sulfuric acid (1 mmol) in 10 mL of ethanol was stirred and refluxed for 7 h. After completion of the reaction, the reaction mass was cooled and ethanol was removed by high vacuum pressure. Then, the reaction mixture was diluted with ethyl acetate and neutralized by bicarbonate solution. The organic layer was separated out and dried over anhydrous sodium sulphate, and concentrated under vacuum; ethyl benzoate was obtained as a white solid. To the ethyl benzoate (**2a**) (1 mmol), hydrazine hydrate (1 mmol) and 15 mL of ethanol were added and refluxed for 5 h. The completion of reaction was monitored by TLC and the traces of hydrazine hydrate were washed with water; acid hydrazide was obtained as a solid. To a mixture of acid hydrazide (**3a**) (1 mmol) and substituted acids (1 mmol), 6 mL of POCl_3_ was added and refluxed at 80 °C for 8 h. The completion of the reaction was monitored by TLC. The reaction mass was quenched with crushed ice and neutralized by potassium carbonate to pH 8. The solid obtained was filtered, washed with water, and subjected to column chromatography (hexane/ethyl acetate eluent)/recrystallized with suitable solvents to obtain pure oxadiazoles. **4a**–**c** (1 mmol), substituted aryl/het.aryl boronic acids (Ar/Het.Ar-B(OH)_2_) (1.2 mmol) and potassium carbonate (3 mmol) were taken in a sealed tube containing H_2_O:dioxane:EtOH (1:5:1) solvent. The above mixture was stirred for 15 min at RT under an inert atmosphere (N_2_ atm). Then, a catalyst (Pd(dppf)Cl_2_) (0.1 mmol) was added to the above reaction mass and heated to 120 °C for 45 min. Aryl-substituted 1,3,4-oxadiazole (**5a**–**x**) was obtained and purified through the chromatographic technique using hexane/ethyl acetate as an eluent.

### 3.2. 2-(3-Bromophenyl)-5-(2,3-dihydrobenzo[b][1,4]dioxin-6-yl)-1,3,4-oxadiazole (***4a***)

Off white solid; 82% yield.

### 3.3. 2-(3-Bromophenyl)-5-(4-methoxybenzyl)-1,3,4-oxadiazole (***4b***)

Off white solid; MP: 68–70 °C; 88% yield; ^1^H NMR (DMSO, 400 MHz): δ 8.01 (s, 1H), 7.89 (d, *J* = 7.6 Hz, 1H), 7.76 (d, *J* = 7.6 Hz, 1H), 7.56–7.42 (m, 1H), 7.26 (d, *J* = 8.4 Hz, 2H), 6.88 (d, *J* = 8.4 Hz, 2H), 4.24 (s, 3H), 3.69 (s, 3H); ^13^C NMR (DMSO, 100 MHz): δ 166.7, 163.3, 158.9, 135.0, 132.1, 130.5 (2C), 129.1, 126.5, 125.9, 125.86, 122.8, 114.6 (2C), 55.5, 30.4; MS: 344.02, *m*/*z* = 345.00 [M + H]^+^, 346.99 [M + 2H]^+^.

### 3.4. 2-(3-Bromophenyl)-5-(3,4-dimethoxybenzyl)-1,3,4-oxadiazole (***4c***)

Off white solid; MP: 74–76 °C; 84% yield; ^1^H NMR (DMSO, 400 MHz): δ 8.04 (s, 1H), 7.92 (d, *J* = 8 Hz, 1H), 7.79 (d, *J* = 8 Hz, 1H), 7.58–7.44 (m, 1H), 6.96 (s, 1H), 6.95–6.79 (m, 2H), 4.24 (s, 2H), 3.71 (s, 3H), 3.69 (s, 3H); ^13^C NMR (DMSO, 100 MHz): δ 166.6, 163.3, 149.3, 148.5, 135.0, 132.1, 129.1, 126.9, 126.0, 125.9, 122.8, 121.4, 113.2, 112.5, 56.0, 30.8; MS: 374.03, *m*/*z* = 375.01 [M + H]^+^, 377.01 [M + 2H]^+^.

### 3.5. 2-(3-(6-Chloro-5-methylpyridin-3-yl)phenyl)-5-(2,3-dihydrobenzo[b][1,4]dioxin-6-yl)-1,3,4-oxadiazole (***5a***)

Off white solid; MP: 164–162 °C; 91% yield; ^1^H NMR (CDCl_3_, 400 MHz): δ 8.50 (s, 1H), 8.28 (s, 1H), 7.82 (s, 1H), 7.72–7.61 (m, 4H), 8.13 (d, *J* = 8 Hz, 1H), 6.99 (d, *J* = 8.8 Hz, 1H), 4.33 (s, 4H), 2.48 (s, 3H); ^13^CNMR (CDCl_3_, 100 MHz): δ 164.6,163.8, 161.2, 146.9, 145.2, 143.9, 137.9, 137.7, 134.6, 132.7, 130.0, 129.9, 126.5, 125.3, 124.9, 120.7, 118.1, 117.9, 116.9, 116.2, 64.6, 64.2, 19.8; MS: 405.09, *m*/*z* = 406.06 [M + H]^+^.

### 3.6. 2-(2,3-Dihydrobenzo[b][1,4]dioxin-6-yl)-5-(4′-(trifluoromethyl)-[1,1′-biphenyl]-3-yl)-1,3,4-oxadiazole (***5b***)

Brown solid; MP: 160–162 °C; 92% yield; ^1^H NMR (CDCl_3_, 400 MHz): δ 8.23–8.19 (m, 2H), 7.96–7.86 (m, 2H), 7.68–7.62 (m, 2H), 7.61–7.44 (m, 4H), 6.97 (d, *J* = 8 Hz, 1H), 4.31–4.28 (m, 4H); ^13^C NMR (CDCl_3_, 100 MHz): δ 164.5, 163.9, 149.1, 148.2, 146.9, 143.9, 138.9, 135.5, 134.6, 130.2, 129.9, 125.4, 124.9, 123.7, 120.7, 118.1, 116.9, 116.2, 64.6, 64.2; MS: 424.10, *m*/*z* = 425.09 [M + H]^+^.

### 3.7. N-Cyclopentyl-3′-(5-(2,3-dihydrobenzo[b][1,4]dioxin-6-yl)-1,3,4-oxadiazol-2-yl)-[1,1′-biphenyl]-3-carboxamide (***5c***)

Off white solid; MP: 186–188 °C; 93% yield; ^1^H NMR (CDCl_3_, 400 MHz): δ 8.26 (s, 1H), 8.05–8.01 (m, 2H), 7.75–7.70 (m, 3H), 7.61–7.59 (m, 2H), 7.50–7.47 (m, 1H), 7.57–7.47 (m, 1H), 6.96 (d, *J* = 8 Hz, 1H), 6.44 (d, *J* = 4 Hz, 1H), 4.45–4.40 (m, 1H), 4.31–4.28 (m, 4H), 2.12–2.02 (m, 2H), 1.75–1.68 (m, 2H), 1.66–1.56 (m, 2H), 1.55–1.52 (m, 2H); ^13^C NMR (CDCl_3_, 100 MHz): δ 167.1, 164.4, 164.0, 146.8, 143.9, 141.3, 140.2, 135.7, 130.3, 129.9, 129.6, 129.1, 126.3, 125.9, 125.8, 125.4, 124.5, 120.7, 118.0, 116.9, 116.2, 64.6, 64.2, 51.9, 33.2, 23.9; MS: 467.18, *m*/*z* = 468.20 [M + H]^+^.

### 3.8. 2-(2,3-Dihydrobenzo[b][1,4]dioxin-6-yl)-5-(3′-methoxy-[1,1′-biphenyl]-3-yl)-1,3,4-oxadia-zole (***5d***)

Off white solid; MP: 118–120 °C; 93% yield; ^1^H NMR (CDCl_3_, 400 MHz): δ 8.32 (s, 1H), 8.08 (d, *J* = 8 Hz, 1H), 7.74 (d, *J* = 8 Hz, 1H), 7.66–7.56 (m, 1H), 7.41–7.73 (m, 1H), 7.25–7.22 (m, 1H), 7.17 (s, 1H), 7.00–6.93 (m, 1H), 4.33–4.32 (m, 4H), 3.88 (s, 3H); ^13^C NMR (CDCl_3_, 100 MHz): δ 164.4, 164.2, 160.1, 146.8, 143.9, 142.1, 141.4, 130.3, 129.9, 129.5, 125.7, 125.5, 124.5, 120.7, 119.7, 118.1, 117.1, 116.2, 113.3, 112.9, 64.6, 64.2, 55.4; MS: 386.12, *m*/*z* = 387.14 [M + H]^+^.

### 3.9. 2-(2,3-Dihydrobenzo[b][1,4]dioxin-6-yl)-5-(3-(naphthalen-1-yl)phenyl)-1,3,4-oxadiazole (***5e***)

Off white solid; MP: 156–158 °C; 97% yield; ^1^H NMR (CDCl_3_, 400 MHz): δ 8.23–8.19 (m, 1H), 7.95–7.86 (m, 3H), 7.67–7.45 (m, 9H), 6.97 (d, *J* = 8 Hz, 1H), 4.30–4.29 (m, 4H); ^13^C NMR (CDCl_3_, 100 MHz): δ 164.4, 164.2, 146.8, 143.9, 141.8, 138.8, 137.6, 133.8, 133.2, 131.4, 129.1, 128.3, 127.1, 126.4, 125.9, 125.8, 125.6, 125.1, 120.7, 118.0, 117.0, 116.2, 64.6, 64.2; MS: 406.13, *m*/*z* = 407.12 [M + H]^+^.

### 3.10. 2-(2,3-Dihydrobenzo[b][1,4]dioxin-6-yl)-5-(3-(pyridin-4-yl)phenyl)-1,3,4-oxadiazole (***5f***)

Brown solid; MP: 194–196 °C; 95% yield; ^1^H NMR (CDCl_3_, 400 MHz): δ 8.71–8.70 (m, 2H), 8.36 (s, 1H), 8.16 (d, *J* = 8 Hz, 1H), 7.78 (d, *J* = 8 Hz, 1H), 7.64–7.61 (m, 3H), 7.58–7.56 (m, 2H), 6.98 (d, *J* = 8 Hz, 1H), 4.32–4.31 (m, 4H); ^13^C NMR (CDCl_3_, 100 MHz): δ 164.5, 163.7, 150.4, 147.1, 146.9, 143.9, 139.2, 129.9, 127.2, 125.3, 124.9, 121.7, 120.7, 118.1, 116.9, 116.2, 64.6, 64.2; MS: 357.11, *m*/*z* = 358.09 [M + H]^+^.

### 3.11. 2-(4-Methoxybenzyl)-5-(3-(pyrimidin-5-yl)phenyl)-1,3,4-oxadiazole (***5g***)

Pink solid; MP: 90–92 °C; 86% yield; ^1^H NMR (CDCl_3,_ 400 MHz): δ 9.24 (s, 1H), 8.99 (s, 2H), 8.21 (s, 1H), 8.09 (d, *J* = 6 Hz, 1H), 7.75–7.59 (m, 2H), 7.28 (d, *J* = 7.2 Hz, 2H), 6.88 (d, *J* = 6.8 Hz, 2H), 4.23 (s, 2H), 3.78 (s, 3H); ^13^C NMR (CDCl_3,_ 100 MHz): δ 165.9, 164.4, 159.1, 158.0, 154.9 (2C), 135.3, 133.2, 130.2, 130.0, 129.9 (2C), 127.2, 125.6, 125.2 (2C), 114.4 (2C), 55.3, 31.1; MS: 344.36, *m*/*z* = 345.08 [M + H]^+^.

### 3.12. 2-(3,4-Dimethoxybenzyl)-5-(3-(pyrimidin-5-yl)phenyl)-1,3,4-oxadiazole (***5h***)

Brown solid; MP: 98–100 °C; 90% yield; ^1^H NMR (CDCl_3,_ 400 MHz): δ 9.24 (s, 1H), 8.99 (s, 2H), 8.22 (s, 1H), 8.08 (d, *J* = 6.8 Hz, 1H), 7.77–7.59 (m, 2H), 6.95–6.78 (m, 3H), 4.23 (s, 2H), 3.85 (s, 6H); ^13^C NMR (CDCl_3,_ 100 MHz): δ 161.2, 159.7, 153.2, 150.2, 144.5, 143.8, 130.5, 128.5, 125.5, 125.3, 124.0, 122.4, 121.2, 120.5, 116.3, 107.2, 106.8, 51.2, 26.8; MS: 374.39, *m*/*z* = 375.09 [M + H]^+^.

### 3.13. 2-(2,3-Dihydrobenzo[b][1,4]dioxin-6-yl)-5-(3-(pyrimidin-5-yl)phenyl)-1,3,4-oxadiazole (***5i***)

Off white solid; MP: 190–192 °C; 91% yield; ^1^H NMR (CDCl_3_, 400 MHz): δ 9.26 (s, 1H), 9.03 (s, 2H), 8.31 (s, 1H), 8.18 (d, *J* = 8 Hz, 1H), 7.74–7.62 (m, 4H), 6.98 (d, *J* = 8 Hz, 1H), 4.32–4.27 (m, 4H); ^13^C NMR (CDCl_3_, 100 MHz): δ 164.6, 163.5, 157.9, 154.9, 146.9, 143.9, 135.4, 133.4, 130.4, 129.9, 127.2, 125.3, 125.2, 120.7, 118.1, 116.8, 116.2, 64.2, 64.0; MS: 358.10, *m*/*z* = 359.11 [M + H]^+^.

### 3.14. 2-(3,4-Dimethoxybenzyl)-5-(3-(4,5-dimethylpyridin-3-yl)phenyl)-1,3,4-oxadiazole (***5j***)

Brown solid; MP: 90–92 °C; 90% yield; ^1^H NMR (CDCl_3,_ 400 MHz): δ 8.56 (s, 1H), 8.19 (s, 1H), 7.97 (d, *J* = 7.6 Hz, 1H), 7.69 (d, *J* = 8.0 Hz, 1H), 7.64 (s, 1H), 7.62–7.48 (m, 1H), 6.96–6.74 (m, 3H), 4.22 (s, 2H), 3.86 (s, 3H), 3.85 (s, 3H), 2.54 (s, 3H), 2.35 (s, 3H); ^13^C NMR (CDCl_3_, 100 MHz): δ 165.6, 164.9, 156.7, 149.3, 148.5, 144.5, 139.0, 135.7, 133.0, 131.6, 130.0, 129.7, 126.1, 125.9, 125.2, 124.6, 121.0, 111.9, 111.5, 55.9, 31.5, 22.2, 19.2; MS: 401.46, *m*/*z* = 402.12 [M + H]^+^.

### 3.15. 2-(2,3-Dihydrobenzo[b][1,4]dioxin-6-yl)-5-(3-(4,5-dimethylpyridin-3-yl)phenyl)-1,3,4-oxadiazole (***5k***)

Off white solid; MP: 198–200 °C; 97% yield; ^1^H NMR (CDCl_3_, 400 MHz): δ 8.60 (s, 1H), 8.29 (_S_, 1H), 8.09 (d, *J* = 8 Hz, 1H), 7.73–7.60 (m, 5H), 6.99 (d, *J* = 8 Hz, 1H), 4.33–4.32 (m, 4H), 2.56 (s, 3H), 2.37 (s, 3H); ^13^C NMR (CDCl_3_, 100 MHz): δ 164.5, 163.9, 156.8, 146.8, 144.6, 143.9, 139.1 135.7, 133.1, 131.6, 129.9, 129.7, 125.9, 125.2, 124.7, 120.7, 118.1, 117.0, 116.2, 64.6, 64.2, 19.2; MS: 385.14, *m*/*z* = 386.15 [M + H]^+^.

### 3.16. 2-(4-Methoxybenzyl)-5-(3-(pyridin-3-yl)phenyl)-1,3,4-oxadiazole (***5l***)

Brown thick mass; 84% yield; ^1^H NMR (CDCl_3,_ 400 MHz): δ 8.85 (s, 1H), 8.61 (s, 1H), 8.20 (s, 1H), 8.06–7.86 (m, 2H), 7.74–7.52 (m, 2H), 7.39 (s, 1H), 7.26 (s, 2H), 6.87 (s, 2H), 4.22, (s, 2H), 3.77 (s, 3H); ^13^C NMR (CDCl_3,_ 100 MHz): δ 165.8, 164.8, 159.0, 148.9, 148.0, 138.7, 135.5, 134.6, 130.2, 129.9 (2C), 126.4, 125.7, 125.4 (2C), 124.8, 123.7, 114.4 (2C), 55.3, 31.1; MS: 343.38, *m*/*z* = 344.11 [M + H]^+^.

### 3.17. 2-(3,4-Dimethoxybenzyl)-5-(3-(pyridin-3-yl)phenyl)-1,3,4-oxadiazole (***5m***)

Yellow solid; MP: 78–80 °C; 88% yield; ^1^H NMR (CDCl_3,_ 400 MHz): δ 8.86 (s, 1H), 8.63 (s, 1H), 8.21 (s, 1H), 8.02 (d, *J* = 7.2 Hz, 1H), 7.91 (d, *J* = 7.6 Hz, 1H), 7.71 (d, *J* = 7.2 Hz, 1H), 7.58 (t, *J* = 7.6 Hz, 1H), 7.40 (d, *J* = 4.8 Hz, 1H), 6.96–6.77 (m, 3H), 4.22 (s, 2H), 3.86 (s, 3H), 3.85 (s, 3H); ^13^C NMR (CDCl_3,_ 100 MHz): δ 165.7, 164.8, 149.3, 149.0, 148.5, 148.1, 138.7, 135.5, 134.5, 130.2, 129.8, 126.4, 126.1, 125.4, 124.8, 123.7, 121.0, 111.9, 111.5, 55.9, 31.5; MS: 373.40, *m*/*z* = 406.12 [M + CH_3_OH + H]^+^.

### 3.18. 2-(2,3-Dihydrobenzo[b][1,4]dioxin-6-yl)-5-(3-(pyridin-3-yl)phenyl)-1,3,4-oxadiazole (***5n***)

Off white solid; MP: 138–140 °C; 95% yield; ^1^H NMR (CDCl_3_, 400 MHz): δ 8.91 (s, 1H), 8.65 (s, 1H), 8.32 (s, 1H), 7.96 (d, *J* = 8 Hz, 1H), 7.74–7.66 (m, 4H), 7.42 (s, 1H), 6.99 (d, *J* = 8 Hz, 1H), 4.33 (s, 4H); ^13^C NMR (CDCl_3_, 100 MHz): δ 164.5, 163.9, 149.1, 148.2, 146.9, 143.9, 138.9, 135.5, 134.6, 130.2, 129.9, 126.4, 125.4, 124.9, 123.7, 120.7, 118.1, 116.9, 116.2, 64.6, 64.2; MS: 357.11, *m*/*z* = 358.09 [M + H]^+^.

### 3.19. 2-(3-(6-Fluoro-5-methylpyridin-3-yl)phenyl)-5-(4-methoxybenzyl)-1,3,4-oxadiazole (***5o***)

Off white solid; MP: 86–88 °C; 91% yield; ^1^H NMR (CDCl_3,_ 400 MHz): δ 8.25 (s, 1H), 8.17 (s, 1H), 8.00 (d, *J* = 7.6 Hz, 1H), 7.82 (d, *J* = 8.8 Hz, 1H), 7.67 (d, *J* = 7.6 Hz, 1H), 7.64–7.50 (m, 1H), 7.29 (d, *J* = 8.4 Hz, 2H), 6.89 (d, *J* = 8.4 Hz, 2H), 4.24 (s, 2H), 3.79 (s, 3H), 2.37 (s, 3H); ^13^C NMR (CDCl_3,_ 100 MHz): δ 165.8, 164.7, 163.3, 159.1, 143.0, 142.8, 140.3, 140.25, 137.9, 133.6, 130.0, 129.9 (2C), 129.8, 126.2, 125.7, 125.3, 124.8, 119.9, 119.6, 114.4 (2C), 55.3, 31.1, 14.5; MS: 375.39, *m*/*z* = 376.11 [M + H]^+^.

### 3.20. 2-(3,4-Dimethoxybenzyl)-5-(3-(6-fluoro-5-methylpyridin-3-yl)phenyl)-1,3,4-oxadiazole (***5p***)

Brown solid; MP: 91–93 °C; 95% yield; ^1^H NMR (CDCl_3,_ 400 MHz): δ 8.14–8.00 (m, 2H), 7.96 (s, 1H), 7.66–7.52 (m, 1H), 7.46 (d, *J* = 7.6 Hz, 1H), 7.07 (d, *J* = 5.2 Hz, 1H), 6.96–6.77 (m, 3H), 4.22 (s, 2H), 3.86 (s, 3H), 3.85 (s, 3H), 2.20 (s, 3H); ^13^C NMR (CDCl_3,_ 100 MHz): δ 165.8, 164.8, 163.3, 149.3, 148.6, 142.9, 142.8, 140.4, 140.3, 137.9, 133.6, 130.9, 130.1, 129.8, 126.2, 126.0, 125.3, 121.0, 120.0, 119.7, 111.9, 111.5, 55.9, 31.5, 14.5; MS: 405.42, *m*/*z* = 406.12 [M + H]^+^.

### 3.21. 2-(2,3-Dihydrobenzo[b][1,4]dioxin-6-yl)-5-(3-(6-fluoro-5-methylpyridin-3-yl)phenyl)-1,3,4-oxadiazole (***5q***)

Off white solid; MP: 144–146 °C; 94% yield; ^1^H NMR (CDCl_3_, 400 MHz): δ 8.27 (d, *J* = 8 Hz, 1H), 8.11 (d, *J* = 8 Hz, 1H), 7.86 (d, *J* = 8 Hz, 1H), 7.70–7.59 (m, 2H), 6.99 (d, *J* = 8 Hz, 1H), 4.33–4.32 (m, 4H); ^13^C NMR (CDCl_3_, 100 MHz): δ 164.5, 163.8, 163.3, 146.9, 143.9, 143.1, 142.9, 140.4, 137.9, 133.7, 130.0, 129.9, 126.2, 125.3, 124.9, 119.7, 118.1, 116.9, 116.2, 115.9, 64.6, 64.2, 14.6; MS: 389.11, *m*/*z* = 390.09 [M + H]^+^.

### 3.22. 2-(3-(2-Fluoro-3-methylpyridin-4-yl)phenyl)-5-(4-methoxybenzyl)-1,3,4-oxadiazole (***5r***)

White solid; MP: 94–96 °C; 87% yield; ^1^H NMR (CDCl_3,_ 400 MHz): δ 8.14–8.01 (m, 2H), 7.96 (s, 1H), 7.67–7.53 (m, 1H), 7.47 (d, *J* = 8 Hz, 1H), 7.28 (d, *J* = 8.8 Hz, 2H), 7.09 (d, *J* =5.2 Hz, 1H), 6.88 (d, *J* = 8.4 Hz, 2H), 4.23 (s, 2H), 3.79 (s, 3H), 2.21 (s, 3H); ^13^C NMR (CDCl_3,_ 100 MHz): δ 166.0, 164.6, 163.9, 159.1, 152.9, 144.2, 144.1, 139.0, 131.7, 129.9 (2C), 129.3, 126.7, 125.5, 124.3, 122.3, 122.28, 117.6, 117.3, 114.4 (2C), 55.3, 31.1, 11.9; MS: 375.39, *m*/*z* = 376.09 [M + H]^+^.

### 3.23. 2-(3,4-Dimethoxybenzyl)-5-(3-(2-fluoro-3-methylpyridin-4-yl)phenyl)-1,3,4-oxadiazole (***5s***)

Brown solid; MP: 84–86 °C; 94% yield; ^1^H NMR (CDCl_3,_ 400 MHz): δ 8.24 (s, 1H), 8.17 (s, 1H), 7.99 (d, *J* = 7.6 Hz, 1H), 7.82 (d, *J* = 8.8 Hz, 1H), 7.67 (d, *J* = 8.4 Hz, 1H), 7.64–7.50 (m, 1H), 6.97–6.78 (m, 3H), 4.23 (s, 2H), 3.86 (s, 3H), 3.85 (s, 3H), 2.36 (s, 3H); ^13^C NMR (CDCl_3,_ 100 MHz): δ 165.8, 164.6, 163.9, 152.8, 149.3, 148.6, 144.3, 144.1, 139.0, 131.7, 130.8, 129.3, 126.7, 126.0, 122.3, 122.26, 121.0, 117.6, 117.3, 111.9, 111.5, 55.9, 31.5, 11.9; MS: 405.42, *m*/*z* = 406.13 [M + H]^+^.

### 3.24. 2-(2,3-Dihydrobenzo[b][1,4]dioxin-6-yl)-5-(3-(2-fluoro-3-methylpyridin-4-yl)phenyl)-1,3,4-oxadiazole (***5t***)

Off white solid; MP: 128–130 °C; 91% yield; ^1^H NMR (CDCl_3_, 400 MHz): δ 8.17 (d, *J* = 8 Hz, 1H), 8.10–8.06 (m, 1H), 7.65–7.60 (m, 3H), 7.49–7.47 (m, 1H), 7.11 (d, *J* = 4 Hz, 1H), 6.99–6.87 (m, 1H), 4.33–4.30 (m, 4H), 2.23 (s, 3H); ^13^C NMR (CDCl_3_, 100 MHz): δ 164.6, 163.9, 163.6, 161.6, 152.9, 146.9, 144.4, 143.9, 139.1, 131.6, 129.4, 129.0, 126.8, 124.6, 122.3, 120.7, 120.5, 118.1, 117.6, 117.3, 116.9, 116.2, 114.0, 64.6, 64.2, 14.1; MS: 389.11, *m*/*z* = 390.13 [M + H]^+^.

### 3.25. 3′-(5-(4-Methoxybenzyl)-1,3,4-oxadiazol-2-yl)-[1,1′-biphenyl]-3-carbaldehyde (***5u***)

Yellow thick mass; 91% yield; ^1^H NMR (CDCl_3,_ 400 MHz): δ 10.09 (s, 1H), 8.26 (s, 1H), 8.12 (s, 1H), 8.01 (d, *J* = 7.6 Hz, 1H), 7.95–7.82 (m, 2H), 7.76 (d, *J* = 7.6 Hz, 1H), 7.70–7.51 (m, 2H), 7.29 (d, *J* = 8.4 Hz, 2H), 6.89 (d, *J* = 8.4 Hz, 2H), 4.24 (s, 2H), 3.78 (s, 3H); ^13^C NMR (CDCl_3,_ 100 MHz): δ 192.1, 165.9, 164.9, 159.0, 140.8, 140.6, 137.0, 133.0, 130.3, 129.9 (2C), 129.72, 129.7, 129.3, 128.0, 126.3, 125.7, 125.4, 124.6, 114.4 (2C), 55.3, 31.1; MS: 370.40, *m*/*z* = 371.19 [M + H]^+^.

### 3.26. 3′-(5-(2,3-Dihydrobenzo[b][1,4]dioxin-6-yl)-1,3,4-oxadiazol-2-yl)-[1,1′-biphenyl]-3-carbal-dehyde (***5v***)

Light yellow solid; MP: 158–160 °C; 94% yield; ^1^H NMR (CDCl_3_, 400 MHz): δ 10.11 (s, 1H), 8.35 (s, 1H), 8.16–8.11 (m, 1H), 7.93–7.90 (m, 1H), 7.80–7.78 (m, 1H), 7.67–7.60 (m, 5H), 7.42–7.38 (m, 1H), 7.00–6.98 (m, 1H), 4.34–4.30 (m, 4H); ^13^C NMR (CDCl_3_, 100 MHz): δ 192.1, 164.5, 163.9, 146.9, 143.9, 140.9, 140.7, 137.0, 133.1, 130.3, 129.8, 129.7, 129.3, 128.2, 126.3, 125.4, 120.5, 118.1, 116.1, 64.6, 64.2; MS: 384.11, *m*/*z* = 385.12 [M + H]^+^.

### 3.27. 1-(5-(3-(5-(4-Methoxybenzyl)-1,3,4-oxadiazol-2-yl)phenyl)thiophen-2-yl)ethanone (***5w***)

Off white solid; 80–82 °C; 90% yield; ^1^H NMR (CDCl_3,_ 400 MHz): δ 8.27 (s, 1H), 7.97 (d, *J* = 6.4 Hz, 1H), 7.76 (d, *J* = 6.4 Hz, 1H), 7.67 (s, 1H), 7.57–7.35 (m, 2H), 7.29 (d, *J* = 6.8 Hz, 2H), 6.89 (d, *J* = 6.8 Hz, 2H), 4.23 (s, 2H), 3.79 (s, 3H), 2.57 (s, 3H); ^13^C NMR (CDCl_3,_ 100 MHz): δ 190.4, 165.9, 164.5, 159.1, 150.8, 143.9, 134.3, 133.3, 129.9, 129.8 (2C), 129.2, 127.0, 125.7, 124.9, 124.8, 124.3, 114.4 (2C), 55.3, 31.1, 26.6; MS: 390.45, *m*/*z* = 391.05 [M + H]^+^.

### 3.28. 1-(5-(3-(5-(3,4-Dimethoxybenzyl)-1,3,4-oxadiazol-2-yl)phenyl)thiophen-2-yl)ethanone (***5x***)

Brown solid; 102–104 °C; 95% yield; ^1^H NMR (CDCl_3,_ 400 MHz): δ 8.26 (s, 1H), 7.98 (s, 1H), 7.81–7.61 (m, 2H), 7.57–7.34 (m, 2H), 6.87 (d, *J* = 12.4 Hz, 3H), 4.23 (s, 2H), 3.86 (s, 6H), 2.56 (s, 3H); ^13^C NMR (CDCl_3,_ 100 MHz): δ 190.5, 165.8, 164.5, 150.7, 149.3, 148.6, 143.9, 134.3, 133.3, 130.8, 129.9, 129.2, 127.0, 126.1, 124.8, 124.3, 121.0, 111.9, 111.5, 55.9, 31.5, 26.6; MS: 420.48, *m*/*z* = 421.06 [M + H]^+^.

### 3.29. MCF-7 and MDA-MB-231 Cell Viability Assay

MCF-7 and MDA-MB-231 (2000) cells were cultured in MEM or Leibovitz’s L-15 medium enriched with 2% FBS and maintained in a humidified atmosphere of 5% CO_2_ at 37 °C [44]. DMSO dissolved compounds were kept as a stock solution, and diluted with a cell culture medium to the desired concentration. Cancer cells (4 × 10^3^) were grown overnight in 96-well plates, cultured, and treated with oxadiazoles or Olaparib at 0, 0.01, 0.1, 10, 100, and 1000 µM concentrations for 72 h. The inhibitory effect of the compounds was assessed using Alamar Blue reagent.

### 3.30. Assay of PARP Activity

PARP activity was measured using the PARP/Apoptosis Universal Colorimetric Assay Kit (4677-096-K, R&D Systems Ltd., Abingdon, UK) [45]. The enzyme inhibition assay uses the histone proteins immobilized 96-well plate, which measures the incorporation of ADP-ribose (biotin-tagged), followed by detection using streptavidin-conjugated horse-radish peroxidase, and finally the color development after the addition of substrate. Oxadiazoles were incubated with the recombinant human PARP enzyme prior to detection.

### 3.31. Western Blot Analysis

Western blot analysis was performed to determine the levels of PARP1 cleavage and p-H2AX in MCF-7 and MDA-MB-231 cells using previously reported methods. Cells at 50–60% confluence were treated with compounds and incubated for 60 h in 2% FBS. MCF-7 and MDA-MB-231 cells were also treated with Olaparib, 5u, or 5s and the levels of cleaved PARP1 and p-H2A-X were determined using the respective antibodies.

### 3.32. Caspase-3 Activity Assay

Caspase-3 activity was measured by Caspase-Glo^®^ 3 assay kit (Promega Madison, WI, USA) according to the manufacturer’s instructions. MCF-7 and MDA-MB-231 cells were seeded in 96-well plates and incubated overnight. Serum-free media or media with compounds were added to cells after 12 h. After a 24h incubation period, Caspase-Glo^®^ 3 reagent and media (1:1) was added into each well and incubated for 30 min in the dark. Luminescence activity (absorbance) was measured using a TECAN reader.

### 3.33. Foci Formation Assay

MCF-7 and MDA-MB-231 cells were plated in six-well plates in 2 mL media with or without compounds and cultured. When foci had formed, MCF-7 and MDA-MB-231 cells were washed with PBS and fixed with ice-cold methanol for 10 min at 4°C. Methanol was removed and crystal violet solution was used to stain foci at room temperature for 10 min. The resultant solution was removed; immobilized MCF-7 and MDA-MB-231 cells were rinsed with distilled water and dried at room temperature.

### 3.34. Molecular Docking Analysis

The molecular docking analysis was performed using the CDOCKER program of Accelrys DS version 2.5 [46]. The co-crystal structures of compound **33** and the PARP1 catalytic domain (PDB ID: 4HHY) were retrieved from RCSB and the binding site of the co-crystal ligand was defined. The protein preparation tool was applied and water was removed from the crystal. The CFF force field was applied to the protein. The lowest energy ring conformation was kept for each compound. We docked compound **33** and compound **5s** (Figure 7) towards the catalytical domain of PARP1 using a docking protocol of the CDOCKER program. A cut-off based on the docking score of reference PARP-1 inhibitors was used, and ligands with the highest score were subjected to visualization using Accelrys graphic interface.

### 3.35. Statistical Analysis

Data were expressed as mean ± SD and statistical analysis was performed using unpaired Student’s *t*-test. A *p*-value of less than 0.05 was considered statistically significant. The graphical presentations were prepared using GraphPad Prism 5 (GraphPad Software, Inc., San Diego, CA, USA). *p* < 0.05 was considered statistically significant, * *p* < 0.05; ** *p* < 0.005.

## 4. Conclusions

In conclusion, newer and diverse structure added-oxadiazoles were designed and synthesized, leading to the identification of structures that produce loss of cell viability in breast cancer cells. Furthermore, the tested oxadiazoles showed dose-dependent inhibition of the catalytic activity of PARP-1 in a cell-free system. Additionally, functional studies of the tested compounds revealed that they may be an initial structure in the drug-development process, wherein inhibition of PARP-1 activity is desirable.

## Data Availability

Not applicable.

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
