# Peer review of "Design and Activity of Novel Oxadiazole Based Compounds That Target Poly(ADP-ribose) Polymerase"

_molecules, 2022, doi:10.3390/molecules27030703_

Round 1
Reviewer 1 Report
The manuscript of Vishwanath and co-workers reports the synthesis and the biological evaluation of novel oxadiazole-based compounds as PARP1 inhibitors.
They synthesized about 25 molecules through a relatively simple synthetic pathway, and then they tested their cytotoxic activity on human breast cancer (MCF-7) and triple negative breast cancer (MDA-MB-231) cell lines. The results indicated that compound 5u and 5s exhibited an IC50 of 1.4 µM and 15.3 µM respectively, and further investigation revealed that both compounds could act as inhibitor of PARP1.
Overall, the work represents an extension of the SAR studies reported so far concerning such inhibitors, so always useful for who is working on (although the compounds are not extremely potent). However, there are some important points that need to be addressed before publication.
- In my opinion the hypothesis on which is based the work is not very well explained. After a brief description of few known compounds showed in Figure 1, the authors declare: “Replacement of the saturated cyclic amine group of many PARP inhibitors such as niraparib, veliparib, E7016, CEP9722 and INO1001 with unsaturated heterocycles such as oxadiazole or triazoles generated a new generation of compounds that are reported to have low nanomolar inhibition of PARP activity and also display robust cellular activity [11-13].” However, the references 11-13 do not contain the fact that the replacement leads to a more potent class of compounds. On the other hand, reference 14 contains a SAR study where the substitution of piperidyl or pyrrolidinyl moieties by several heterocycles was carried out, representing the correct reference that need to be cited. So, I suggest to modify the paragraph and the cited references in order to avoid any misunderstanding.
- More important, in the reference 14 (Journal of Medicinal Chemistry, 2009, Vol. 52, No. 21) is reported (quote): “all the compounds synthesized were potent against PARP-1, since the heterocycle portion of the inhibitors was generally not involved in any key interactions with the enzyme according to X-ray cocrystal structures from early studies.” I am therefore wondering on which bases the herein authors decided to use the oxadiazole as main scaffold for the design of new PURP inhibitors.
- I would also suggest making 2 different figures in the same paragraph: one with the FDA approved drug and one with the last 3 compounds to make easier the reading. For the same reasons, I would also delete the IUPAC name of the compounds within the text.
- Line 81: Please delete “in continuation [17-26]”. Those references are not really centered to the topic, so I do not understand why this work represent a continuous, but mostly self-citations…
- There are not yields reported for the synthetic steps in Scheme 1 Please insert them, at least as range.
- Can you comment more deeply the Figure 2 (efficacy of 5u and 5s in comparison to Olaparib)?
- Why have you not tested the binding affinity of your compounds on the PURP1 enzyme? It would bring more value to the work.
- Line 122: please add a reference consistent to what you are saying in line 122.
- In silico analysis: what is compound 33? For what I am understanding, you are doing a superimposition, right? It is not very well explained. I suggest to re-write the initial part of the paragraph. Is the oxadiazole ring participating on the interaction then?
- The conclusions are a bit too short and not very conclusive.
- Line 211: What do you mean with “vacuum high pressure”?
- Experimental part: Eluent of TLC and/or Column chromatography are never mentioned.
- MS analysis are missing for compounds 5g, 5h, 5j, 5l, 5m, 5o, 5p, 5r, 5s, 5w and 5x.
- The NMR spectra are not always very clean or well resolved, but this can happen with difficult compounds. However, the purity can be an issue if the compound is the evaluated for any biological activity. Thus, I have some perplexities regarding the general purity of the final products. I also find some inconsistencies when I compare some not very clean or resolved NMR spectras (example: 5q) with the melting point range (2 °C, so pretty pure compounds). Perhaps the insertion of the whole LC-MS analysis can help to clarify my concerns.
- Can you comment on the double MS (peak m/z 741) present in the MS spectra of 5u?
- Several typos are present:
- Line 37: a double blank is present between “determined” and “that”.
- Line 71: a double blank is present between “observed” and “to”.
- Line 69: Compound 2 should be Compound 1.
- Throughout all manuscript, the numbering of the compounds in the text and in the figures should be in bold.
- Line 72: a double blank is present between “EC50” and “of”.
- Line 72: a double blank is present between “an” and “excellent”.
- Line 82: a double blank is present between “of” and “the”.
- In the biological assays part, all mM should be change to “microM” (µM).
- Figure 2 caption: a blank is missing between 560 and nm.
- Line 136: a double blank is present between “again” and “similar”.
- Material and Methods: 1H and 13C should be written as 1H and 13C
- CDCl3 should be written as CDCl3
Author Response
Plz see the attached file

Reviewer 2 Report
The authors describe the synthesis of twenty-four oxadiazoles and their antiproliferative activity against two breast cancer cell lines. Although IC50 values are generally only mediocre, some compounds show cytotoxicity comparable to olaparib and interesting effects on PARP1 and caspase 3 expression.
Nonetheless, the authors leave some important questions unresolved:
- With which design model did the authors select the syntones with which they decorate the three oxadiazole scaffolds? In other words, were the aromatic rings inserted via Suzuki coupling selected to explore a specific chemical diversity?
- Are the authors able to explain why “all of the compounds which bear a pyridine-ring produced significant loss of cell viability”?
- Are the authors able to explain why “compound 5u exhibited an IC50 of 1.4 mM which could be attributed to the presence of naked benzaldehyde group”?
- Could the significant variation in lipophilicity play a decisive role in this regard?
I believe that a more exhaustive discussion of structure-activity relationships is needed.
Minor remarks
|
Line(s) |
text |
suggested action |
|
35 |
BRCA1 and BRCA2 |
BReast CAncer gene 1/2 (BRCA1 and BRCA2) |
|
Figure 1 |
olaparib |
check the structure |
|
Figure 1 |
caption |
review the test |
|
60-69 |
---- |
too long sentence |
|
69-70 |
compounds 2 (Figure 1) |
compound 1 (Figure 1) |
|
the IUPAC name is not required |
||
|
73-74 |
called G007-LK |
called G007-LK (Figure 1) |
|
---- |
the IUPAC name is not required |
|
|
78-79 |
compound 2 |
compounds 2 (Figure 1) |
|
---- |
the IUPAC name is not required |
|
|
81 |
synergized with Olaparib |
In combination with Olaparib |
|
82 |
subsitutions of the pyridine, biphenylic and benzodiaxazole rings |
substitutions of the pyridine, biphenyl and (?) rings |
|
106 |
23.1 mM, |
23.1 mM, |
|
108 |
IC50 of 1.4 mM |
IC50 of 1.4 mM |
|
114 |
1, 10, 100 mM |
1, 10, 100 mM |
|
134 |
prototypical |
unsuitable adjective |
|
141 |
(1 or 10 mM). |
(1 or 10 mM). |
|
143 |
Activated CPP32 is a prototypical caspase that becomes activated during apoptosis |
definition already reported (see line 134) |
|
149 |
at 1 mM or 10 mM concentration |
at 1 mM or 10 mM concentration |
|
171 |
Olaprib (1 or 10 mM), |
Olaprib (1 or 10 mM), |
|
176-178 |
compound 33 |
insert the structure of the compound 33 |
|
---- |
the IUPAC name is not required |
|
|
180-184 |
The analysis of the results indicated that the compound 33 core region called benzo[de][1,7]naphthyridin-7-(8H)-one was occupying the active site of the catalytic domain of PARP1, which was similar 183 when compared to the phenylic-pyridine group of active compound 5s, that showed high 184 negative CDOCKER energy. |
Reformulate this sentence |
|
192 |
Pi-Pi |
p-p |
|
204-206 |
1H and 13C NMR |
1H and 13C NMR |
|
CDCl3 |
CDCl3 |
|
|
208 |
General procedure for the synthesis of Suzuki coupled 2,5-disubstituted-1,3,4-oxadiazole. |
General procedure for the synthesis of 2,5-disubstituted-1,3,4-oxadiazole. |
|
235 |
[M+ 1]+ |
[M+ H]+ also correct in all subsequent sections |
|
sections |
3.8.2.; 3.9.2.; 3.11.2; 3.13.2; 3.14.2; 3.16.2; 3.17.2; 3.19.2; 3.20.2; 3.24.2; 3.25.2 |
the m/z value is missing |
The twenty-four compounds listed in Table 1 can be presented more clearly and in less space. For example, it is possible to identify, in a new more compact table, the three scaffolds on which only the aromatic substituent carried by the corresponding boronic acid varies.
Reviewer 3 Report
The manuscript quoted molecules-1436382 and titled "Design and activity of novel oxadiazole based compounds that 2 target Poly(ADP-Ribose) Polymerase" by Vishwanath et al, reports the synthesis of a new class of oxadiazole-based ligands that are predicted to target PARP1. The manuscript is clearly presented and the data are consistent. Screening of various ligands by loss of cell viability in mammary carcinoma cells revealed that seven compounds 18 possessed IC50 values in the range of 1.4 to 25 μM. Furthermore, the compound 5u inhibited the viability of mammary carcinoma cells (MCF-7) with an IC50 value of 1.4 µM, when compared to Olaparib. This manuscript is therefore acceptable for publication in Molecules after the implementation of the following issues.
- The purity of the final compounds should be indicated by HPLC and the same should be attached with the manuscript. I did not find any HRMS data or elemental analyses of the compounds, it must be done to check whether compounds are > 99% pure to validate the screening results.
- Abstract should be shortened, make it more emphatic and novel word from title should be removed otherwise authors should define the novelty of such molecules.
- The spectroscopic characterization of the intermediates should also be reported in the experimental procedures in the supplementary files. For known compounds adequate reference to previous reports should be included.
- The structures in Table 1 should be presented in a different way - they cannot be presented on 5 pages of the manuscript.
- I also cannot find the relevant statistic for the IC50 values in Table 1 (standard error of the mean or confidence interval). The Authors should then calculate them.
- Conclusions should be discussed in more detail.
Round 2
Reviewer 1 Report
The manuscript has been improved and, in my opinion, it could be ready for publication.
A couple of further comments:
a) I cannot see the structure of compound 33;
b) several typos are still present in the manuscript;
c) the supporting information's layout is a bit messy and therefore difficult to read.
Author Response
We thank the referee for his valuable comments and we revised them accordingly.
“The manuscript has been improved and, in my opinion, it could be ready for publication.
A couple of further comments:
- I cannot see the structure of compound 33;
Our response: Added the structure appropriately
- several typos are still present in the manuscript;
Our response: Revised accordingly.
- the supporting information's layout is a bit messy and therefore difficult to read.
Our response: corrected now.

Reviewer 2 Report
I believe that the revised version of the manuscript Design and activity of novel oxadiazole based compounds that target Poly(ADP-Ribose) Polymerase by Divakar Vishwanath et al., responded to all the most important remarks. For this reason I think I recommend its publication on Molecules.
Author Response
We thank the reviewer for his valuable suggestions to improve our manuscript.
